# DUSP9, a Dual-Specificity Phosphatase with a Key Role in Cell Biology and Human Diseases

**DOI:** 10.3390/ijms222111538

**Published:** 2021-10-26

**Authors:** Fatma Zohra Khoubai, Christophe F. Grosset

**Affiliations:** INSERM, MIRCADE Team, UMR1035 Biothérapie des Maladies Génétiques, Désordres Inflammatoires et Cancer, BMGIC, Université de Bordeaux, 146 Rue Léo Saignat, F-33000 Bordeaux, France; fatma-zohra.khoubai@u-bordeaux.fr

**Keywords:** mitogen-activated protein kinase, dual-specificity phosphatase, MAP kinase phosphatase, sex differences, metabolic syndromes, cancer, therapy

## Abstract

Mitogen-activated protein kinases (MAPKs) are essential for proper cell functioning as they regulate many molecular effectors. Careful regulation of MAPKs is therefore required to avoid MAPK pathway dysfunctions and pathologies. The mammalian genome encodes about 200 phosphatases, many of which dephosphorylate the MAPKs and bring them back to an inactive state. In this review, we focus on the normal and pathological functions of dual-specificity phosphatase 9 (*DUSP9*)/MAP kinase phosphatases-4 (MKP-4). This cytoplasmic phosphatase, which belongs to the threonine/tyrosine dual-specific phosphatase family and was first described in 1997, is known to dephosphorylate ERK1/2, p38, JNK and ASK1, and thereby to control various MAPK pathway cascades. As a consequence, *DUSP9* plays a major role in human pathologies and more specifically in cardiac dysfunction, liver metabolic syndromes, diabetes, obesity and cancer including drug response and cell stemness. Here, we recapitulate the mechanism of action of *DUSP9* in the cell, its levels of regulation and its roles in the most frequent human diseases, and discuss its potential as a therapeutic target.

## 1. General Introduction

The mitogen-activated protein kinase (MAPK) signaling pathways are crucial in cell function and homeostasis. MAPKs regulate pathophysiological processes by controlling signal translation and cellular response such as survival, proliferation, differentiation and migration [1,2,3]. They are activated by a double phosphorylation process on tyrosine and threonine residues in a conserved Thr-X-Tyr motif (X being any amino acid) [4,5]. Activation of MAPK pathways triggers multiple intracellular signaling cascades. Each cascade is initiated by a specific signal and leads to the activation of a particular MAPK [6]. Once activated, MAPK can phosphorylate various cytoplasmic and/or nuclear substrates and induce changes in the function of target proteins and gene expression [6]. Spatial localization of MAPKs also determines the target substrates and subsequent cellular effects [7].

In humans, MAPKs are divided into three main families: ERK (kinases regulated by the extracellular signal), p38 (protein kinases activated by stress) and JNK (amino-terminal kinases Jun) [2,3,6]. Each of the MAPK modules involves three sequentially activated kinases: MAPKKK, MAPKK and MAPK [4,5,8]. ERK is the best-studied member of the MAPKs. It is activated by numerous extracellular signals such as growth factors, mitogens, cytokines, viruses, receptor ligands coupled to G proteins, and oncogenes. It thus regulates cell signaling under normal and pathological conditions. When the MAPK pathway is inactive, ERK is localized in the cytoplasm. Once activated, phosphorylated ERK is dimerized and is translocated to the nucleus [4,8]. In the nucleus, phosphorylated ERK binds to transcription factors such as proto-oncogenes c-Fos and c-Jun and induces the expression of many genes, most of which are associated with cell proliferation, motility, stemness and survival [8]. As a consequence, MAPK signaling and ERK kinases play a key role in cancer by regulating proliferation, migration, angiogenesis and metastasis [8]. ERK signaling is deregulated in about a third of human cancers and constitutive ERK activation has been reported in many types of tumors [9]. The ERK cascade involves several upstream kinases including RAS, RAF and MEK1/2, forming the RAS/RAF/MEK/ERK pathway and several downstream MAPKs [8,10]. RAS/RAF/MEK/ERK signaling dysfunction is a major trigger for the development of several cancers and RAS mutations (mainly K-RAS) account for 30% of all analyzed tumors [8,11]. Regarding p38 and JNK, they are mainly linked to stress and cell apoptosis [8]. There are four p38 MAP kinases in mammals: α, β, γ and δ. Of all the p38 MAPK isoforms, p38a is the best characterized and is expressed in most cell types [4].

According to the PhosphoSite database, approximately 17,000 proteins have at least one phosphorylated residue in their peptide sequence [12]. Therefore, very precise regulation of protein phosphorylation is necessary for the proper functioning of cells. Deregulation of MAPKs has been described in various pathologies such as metabolic diseases and malignancies in association with drug resistance [2,3,7,12,13,14]. Under normal conditions, MAPKs are tightly regulated by protein phosphatases. Phosphatases reverse phosphorylation and return MAPKs to an inactive state. The dual specificity phosphatase (DUSP) family belongs to the 199 phosphatases encoded in the human genome. This family is composed of 61 phosphatases capable of downregulating MAPKs by dephosphorylating both tyrosine and serine/threonine residues in a single substrate [12,15]. The phosphorylation of proteins is a reversible process. This prevents the abnormal activation of the signal and fine-tunes its activity and downstream effects [3,12]. The balance between phosphorylation and dephosphorylation controls the expression, function, activity and localization of many proteins [12,16]. Dephosphorylation by DUSPs regulates the duration, intensity and spatiotemporal profile of the MAPK signaling cascade [17]. This dephosphorylation takes place thanks to the highly conserved phosphatase site which contains arginine, cysteine and aspartic acid [3,12,18]. In addition to the active site common to all DUSPs, some DUSPs contain a MAP kinase-binding motif (MKB), also called a kinase-interacting motif (KIM), which interacts with the common docking domain of MAPKs to allow the interaction between the enzyme and the substrate [3,12,18,19]. Ten DUSPs containing the KIM domain are classified as typical DUSPs or MAP kinase phosphatases (MKPs) (Table 1), while those which do not have this domain (16 phosphatases in total) are called atypical DUSPs [3,5,20]. However, there are a few exceptions. *DUSP2*, *DUSP5* and *DUSP8* are typical DUSPs and contain the KIM domain but they are not called MKPs. On the other hand, *DUSP14* and *DUSP26*, which are atypical DUSPs and do not contain a KIM domain, are called MKP6 and MKP8, respectively (Table 1) [3,20]. Typical DUSPs are the best characterized within the DUSP family [20] and this comprises the typical DUSP named *DUSP9* or MKP4, which was first described in 1997 by Muda and collaborators [18]. This 42-kDa protein dephosphorylates several substrates including JNK, p38, the MAPKKK apoptosis signal-regulating kinase 1 (ASK1) and ERK1/2 with a high specificity for ERK kinases [3,18]. In this review, we discuss recent findings about *DUSP9*, a cytoplasmic phosphatase encoded by a four-exon gene located on the X chromosome [18,21].

## 2. General Characteristics of *DUSP9* and Mechanisms of Regulation

*DUSP9* is a typical DUSP characterized by the presence of an MKB/KIM motif and a phosphatase domain, which shares structural homology with other DUSPs [12]. Sequence homology analysis of *DUSP9* showed 61% identity with *DUSP22*/MKP-X, 57% with *DUSP6*/MKP-3 and 35% with *DUSP8* [18]. *DUSP9* contains a C-terminal catalytic domain common to all DUSPs. The core of this domain, which consists in residues of arginine, cysteine and aspartic acid (Figure 1), is highly conserved and carries the phosphatase activity [11,18,19,43]. Arginine at position 296 forms hydrogen bonds with a phosphate group on the substrate and stabilizes the transition state. Cysteine at position 290 functions as an active nucleophile site forming a covalent thiol-phosphate intermediate, while aspartic acid in position 259 acts as a catalytic acid to give a proton to the leaving group (Figure 1) [18,43,44]. 

The MKB/KIM motif is composed of two CDC25 homology domains and an intermediate group of basic amino acids mediating the interaction with the common domain of MAPKs [16,18,19,44]. *DUSP9* is able to undergo a conformational rearrangement allowing it to alternate between a partially active structure and a fully active structure [43]. A crystallographic representation of the catalytic site of *DUSP9* showed a unique structure with significant differences between the catalytic core and several surrounding loops compared to other MKPs. The catalytic site of *DUSP9* deviates considerably from the canonical conformation of DUSPs, which may explain the low catalytic activity of this protein in the absence of specific substrates [19,43]. The *DUSP9* protein alone has very low catalytic activity but binding to MAPK through the MKB/KIM domain significantly increases its phosphatase activity [18,19]. The binding of the substrate likely triggers a conformational change and thus increases its catalytic efficiency [19,43]. *DUSP9* is capable of binding and being activated by different MAPKs. Measurements in the presence of para-nitrophenylphosphate (pNPP) showed *DUSP9* binding preference and activation by ERK2, JNK and p38 MAPK [19,34,44].

At a functional level, *DUSP9* phosphatase is unique. It plays an important role in the dephosphorylation and inactivation of specific kinases such as ASK1, ERK1/2, p38 and JNK (Figure 2). This results in a negative control of MAPK signal transduction and a fine tuning of their duration and intensity [15,17,32]. The endogenous catalytic activity of *DUSP9* was studied for the first time by Muda and collaborators by measuring the hydrolysis of pNPP in the presence of increasing doses of a purified human recombinant protein. *DUSP9* displayed a dose-dependent catalytic activity which was directly proportional to the amount of protein added [18]. The *DUSP9* protein has a broad specificity for MAPK substrates. It can dephosphorylate ERK-family MAPKs, stress-activated JNK and p38 MAPKs (Figure 2), but its effect is significantly higher and more specific for ERK kinases [18,43]. In hepatic tumor cells, *DUSP9* negatively regulates the RAS/RAF/MEK/ERK signal by dephosphorylating ERK1/ERK2 and a low level of *DUSP9* is correlated with an elevated level of phospho-ERK1/2 in hepatocellular carcinoma (HCC) samples [15]. Co-incubation of ERK2 with increasing concentrations of *DUSP9* results in the dose-dependent blockade of ERK2 target phosphorylation such as stathmin [18]. This catalytic activity is effectively inhibited by sodium vanadate, which is an inhibitor of protein tyrosine phosphatases [18].

In normal mature tissues, *DUSP9* is mainly expressed in kidney, adipose tissue and placenta, while it is only minimally present in brain, ovary, testis and urinary bladder (National Center for Biotechnology Information: available online: https://www.ncbi.nlm.nih.gov/gene/1852, accessed on 15 July 2021) [32,46,47]. In order to maintain cell homeostasis, the level and activity of *DUSP9* has to be tightly regulated at transcriptional, post-transcriptional and post-translational levels (Figure 2) [3]. At the transcriptional level, the expression of *DUSP9* can be regulated by various transcription factors such as EFS family members and hypoxia-inducible factor 1 alpha (HIF1α). For instance, genetic depletion by gene knock-down or digoxin-induced pharmacological inhibition of *HIF1* blocks the expression of *DUSP9* and induces the loss of its inhibitory effect on the ERK signaling pathway in MDA-MB-231 breast cancer cells [17]. Thus, *HIF1* positively regulates *DUSP9* expression. *DUSP9* is also transcriptionally regulated by BMP signaling in mouse embryonic stem cells (mESCs). The bone morphogenetic protein 4 (Bmp4) induces the recruitment of Smad1/5 and Smad4 on the promoter region of *DUSP9* and induces *DUSP9* expression at the mRNA level, and thus at the protein level. The positive regulation of *DUSP9* by Bmp4 is accompanied by inhibition of ERK pathway activity and downregulation of its targets Egr1 and Fos (Figure 2). This *DUSP9*-mediated dephosphorylation was not observed for p38 and JNK, showing its specificity for ERK1/2 signaling which is crucial for the renewal and differentiation of mESCs [31]. In the same cells, *DUSP9* can also undergo post-translational regulation by a long non-coding RNA (lncRNA) called *LincU* [48]. Following induced expression of *LincU* by the transcription factor Nanog, this lncRNA directly binds *DUSP9* and maintains the phosphatase in an active and stable conformation. As a result, *DUSP9*, bound to *LincU*, constantly dephosphorylates ERK1/2 and thus, totally blocks RAS/RAF/MEK/ERK signaling. Interestingly, the phosphorylation level of ERK1/2 was inversely correlated with *LincU* expression and the amount of *DUSP9* protein, while the level of *DUSP9* mRNA remained unchanged [48]. Thus, *LincU* interacts with *DUSP9* protein and stabilizes it in an active state, thereby triggering a constitutive dephosphorylation state of *DUSP9* kinase substrates (Figure 2) [48]. *DUSP9* can also be transcriptionally regulated by the retinoic acid receptor (RAR). Microarray analysis of differential gene expression in Caco-2 cells treated with RAR agonists, Ch55 and Am580 demonstrated dose-dependent induction of *DUSP9* expression by RAR signaling. *DUSP9* was one of the most induced genes after RAR activation. Induction of *DUSP9* mRNA and protein by RAR signaling was confirmed by RT-qPCR and immunolabelling [49]. Similar results were obtained with HT29 and HeLa cells. On the other hand, *DUSP9* induction was inhibited in Caco-2 cells expressing the dominant negative form of RARα or treated by the specific RAR antagonist LE540. Chromatin immunoprecipitation analysis showed that RAR induces *DUSP9* expression by binding directly to the *DUSP9* promoter through an inverted direct repeat separated by 1 (DR1 element). By inducing *DUSP9*, RAR signaling inactivates ERK during the differentiation of colorectal cancer (CRC) cells [49]. In a more recent paper, an upregulation of *DUSP9* was also reported in rat ovaries following Estrogen Receptor-beta signaling induction, supporting the hypothesis that *DUSP9* is one of the key genes involved in gonadotrophin-mediated ovarian follicle development [50].

In the cytoplasm, *DUSP9* can be degraded by the proteasome following its ubiquitination [3,48]. Pretreatment with the potent proteasome inhibitor MG132 stabilizes *DUSP9* protein and increases its level in *LincU*-deficient mESCs [48]. These data indicate that the ubiquitination-proteasome pathway is involved in the degradation of *DUSP9* induced by *LincU* deficiency. Immunoprecipitation performed after pretreatment with MG132 and immunolabelling with an anti-ubiquitin antibody showed polyubiquitinated *DUSP9* bands, suggesting that the knockdown of *LincU* causes ubiquitination of *DUSP9* [48]. This indicates that *LincU* binds to *DUSP9*, increases its stability and protects it from ubiquitin-mediated degradation [3,48].

Similar findings were obtained in human embryonic stem cells (hESCs). The overexpression of *LincU* in hESCs also protected *DUSP9* from degradation and inhibited the phosphorylation of ERK1/2, indicating the conserved role of *LincU* in the stabilization of *DUSP9* [48]. Finally, *DUSP9* can be regulated by microRNAs (miRNAs), which are small non-coding RNAs that act as post-transcriptional regulators and which are often deregulated in pathological conditions and cancers [10,51,52,53,54]. MiR-1246 and miR-212 can target the 3′-untranslated region (UTR) of the *DUSP9* transcript and reduce its expression in CRC cells and hepatoblastoma-derived HepG2 cells, respectively [55,56]. Moreover, Chang and collaborators previously reported a link between *DUSP9* expression and miR-133b and miR-4458 in CRC [57]. In our data, we noticed a decreased level of *DUSP9* protein following ectopic expression of *miR-4510* in hepatoma cells Huh7 [10].

## 3. DUSP9 in Embryonic Stem Cell Pluripotency and Sex Differences

The feeder-free culture of undifferentiated mESCs depends on the cumulative effect of pluripotency factors (i.e., LIF, BMP4) and low activity of commitment-associated pathways, including the MAPK pathway. In 2012, Li and collaborators investigated the role of *DUSP9* in cell stemness and pluripotency [31]. They found that, associated to Lif, Bmp4 supports the pluripotency state of mESCs by keeping ERK pathway activity at a low level, independently of the upstream MAPKK MEK kinase. As Bmp4 is strongly connected to Smad signaling, they showed that Bmp4 mediates its inhibitory effect on ERK by inducing the binding of transcriptional Smad1/5+Smad4 complex on the *Dusp9* promoter, thereby promoting its transcription and expression. An increase in *DUSP9* led to the dephosphorylation of ERK1/2 and the downregulation of the ERK pathway target genes *Egr1* and *Fos*. By comparison, the Bmp4/Smad/*DUSP9* axis was inactive in murine somatic cells [31]. Altogether, these data demonstrated that *DUSP9* is an important downstream regulator of BMP/Smad signaling and a key factor in maintaining the stemness of mESCs.

The pluripotency of embryonic stem cell lines is also closely associated with the DNA methylation status of the cells. In an attempt to shed light on this link, Choi and collaborators generated isogenic models of male and female mESCs and murine embryonic germ cells (mEGC), and found that sex rather than embryonic cell type determines the DNA methylation status [58]. The methylation pattern in these undifferentiated cells was controlled by the ratio of X chromosomes to autosomes. Comparative transcriptomic analysis in male and female isogenic embryonic stem cell lines identified *DUSP9* as an X chromosome-linked gene more abundantly expressed in female than in male mESCs. In line with the lower activity of MAPK pathway in female mESCs compared to male ones and the presence of *ERK1/2*, *p38* and *Jnk1/2/3* genes to autosomes, the methylation level of DNA was reduced in cells ectopically expressing *DUSP9* [58]. Forced expression of *DUSP9* reduced the global methylation levels in male mESCs, while its deletion in female ones led, after several passages, to an elevation of methylation levels. In *DUSP9*-overexpressing male cells, the MAPK pathway was inhibited and its downstream targets were less expressed, while the naïve pluripotency markers Ror2 and Prdm14 were increased and the primed pluripotency marker Dmnt3B was decreased [58]. These data demonstrated that *DUSP9* plays an important role in embryonic stem cell sex differences and methylation status. In female cells, X chromosome-linked *DUSP9* expression keeps the MAPK pathway activity and the expression of *DNMT3A/3B* at a low level, leading to a global hypomethylation by comparison to male cells [58]. These results were corroborated by a more recent study in which the authors showed that the effect of *DUSP9* on global hypomethylation and sex differences was uncoupled from the opening of the chromatin, cell growth and the delay to exit pluripotency [59].

In a more recent study using an elaborate multistep screening approach and mESCs, the authors found that *DUSP9* is an X-chromosomal gene and is involved in sex differences during early development [60]. In vivo, its expression was higher in females than in males. Forced expression of *DUSP9* enhanced the pluripotency of mESCs by dephosphorylating Mek and ERK1/2, thereby decreasing the expression of the pluripotent factors *NANOG* and *PRDM14*. The study also confirmed the role of *DUSP9* in the regulation of global DNA methylation mediated by MAPK pathways [60].

## 4. DUSP9 and Metabolic Diseases

Insulin resistance is defined as the failure of many cells to respond to insulin hormone, to mediate insulin signaling and to take up glucose, thereby leading to abnormally high blood sugar levels known as hyperglycemia. This metabolic syndrome is often associated with chronic inflammation, cell stress responses and pathological conditions such as obesity, non-alcoholic fatty liver disease (NAFLD) and type 2 diabetes mellitus [34,47,61]. Molecularly, the normal response of cells to insulin stimulus depends on tyrosine-kinase insulin receptor membrane expression and downstream phosphorylation cascades and effectors, which comprise the pro-mitotic ERK1/2-dependent MAPK pathway, the JNK- and p38 MAPK-dependent SAP pathway response, the pro-metabolic PI3K/PDK1/AKT-pathway and also the fuel-sensing enzyme AMP-activated protein kinase (AMPK) [61,62,63,64]. Thus, insulin resistance is a pathological condition associated with over-activated phosphorylation cascades involving MAPK, PI3K, AMPK and SAP kinases.

The first report of a link between *DUSP9* and insulin resistance was published in 2003 [47]. Using an alkaline-phosphatase-based functional screen reproducing the insulin-operated transcriptional shut down of PEPCK gene in target cells, Xu and collaborators identified *DUSP9* out of 10,000 cDNA clones as a potent inhibitor of insulin signal transduction [47]. In physiological conditions, *DUSP9* is expressed in mature brain, kidney, testis, white adipose tissue, embryonic liver and placental trophoblast giant cells [18,46,47,65]. By comparing the expression of *DUSP9* in different murine models, Xu and collaborators found that it becomes detectable in brown adipose tissue, liver and muscle of obese mice, while being undetectable in these tissues in mice fed on a regular diet [47]. *DUSP9* expression remained unchanged in kidneys and testis in the different tested murine models. Next, they showed that *DUSP9* expression increases during insulin-induced adipogenesis and culminates in mature adipocytes. Interestingly, forced *DUSP9* expression impedes insulin-mediated adipocyte differentiation and limits the uptake of glucose in mature adipocytes [47]. Overall, these data identified *DUSP9* as a key regulator of insulin signaling and highlighted its potential role in insulin resistance and metabolic diseases by dephosphorylating kinases involved in metabolic processes, glucose uptake and storage.

In 2004, Bazuine and collaborators demonstrated the negative role of *DUSP9* in induced insulin resistance through an alternative phosphorylation cascade governed by p38 MAPK [66]. They also showed that dexamethasone, which inhibits insulin signaling and lowers glucose uptake, induces *DUSP9* and *DUSP1/MKP-1* expression in adipocytes through a direct or indirect mechanism involving the glucocorticoid receptor. Here again, the expression of *DUSP9* decreased glucose uptake and led to ERK dephosphorylation, mimicking the action of dexamethasone. However, unlike Xu and collaborators [47], Bazuine and collaborators showed that *DUSP9* attenuates arsenite-mediated but not insulin-mediated p38 MAPK dephosphorylation [66]. These discrepancies can be explained by the pleiotropic effect of *DUSP9* on multiple signaling pathways, as it can dephosphorylate and inactivate either ERK1/2, JNK, p38 and/or ASK1 kinases depending on the tissue and biological conditions [34,66,67]. Nevertheless, the insulin receptor substrate-1 (IRS1), which is the target of many of these kinases [61], is phosphorylated on Ser-307. This phosphorylation blocks the subsequent tyrosine phosphorylation of IRS1 by the kinase domain of the insulin receptor after insulin binding. Once phosphorylated on ser-307, IRS-1 dissociates from p85 PI3K and impairs downstream PDK1/AKT signaling [67]. By dephosphorylating and inactivating ERK1/2, JNK, p38 MAPK and/or ASK1 kinases, *DUSP9* restores the tyrosine phosphorylation level of IRS-1 and its capacity to interact with p85-PI3K and mediate insulin signal transduction. Moreover, *DUSP9* can also impair the action of extracellular mediators and stress inducers (i.e., proinflammatory cytokines, oxidative compounds, endoplasmic reticulum stress effectors), which can induce insulin resistance by abnormally activating MAPK or SAP pathways [34,67]. Importantly, *DUSP9* can impede insulin resistance in vivo in the *ob/ob* murine model by lowering blood glucose levels and by restoring the normal level of enzymes involved in gluconeogenesis and lipogenesis such as fructose-1,6-biphosphatase, SREBP1C, SCD1 and ACC [34,67]. As a result, hyperglycemia, the synthesis of lipids and the accumulation of triglycerides in liver decrease, thereby attenuates the impact of overall hepatic steatosis. On the other hand, *DUSP9* expression can be modulated in insulin-sensitive tissues such as white adipocyte tissue, liver and muscle in normal or pathological conditions, depending on both the type of feeding diet and tissue [67]. 

In agreement with the two previous studies, Ye and collaborators reported the protective effect of the ectopic expression of *DUSP9* against nonalcoholic steatohepatitis (NASH), the most frequent and pathophysiological form of NAFLD [34]. NASH is usually associated with type 2 diabetes and obesity, and is characterized by chronic inflammation, insulin resistance and lipid accumulation. Using liver-specific murine models, various feeding conditions and genetic gain- and loss-of-function approaches, they showed that the *DUSP9* protein level is slowly and steadily decreased in the liver of mice fed on a high fat diet or presenting obesity. Interestingly, this insulin resistance-induced down-regulation of *DUSP9* occurred at the protein level but not at the transcript level, suggesting the translational regulation and/or the potential implication of miRNAs [34]. Interestingly, we found a significant down-regulation of *DUSP9* in HCC-derived Huh7 cells transfected with miR-4510, a primate-specific miRNA which inactivates Wnt/β-catenin and RAS/RAF/MEK/ERK signals by targeting RAF proto-oncogene serine/threonine-protein kinase RAF1 and glypican-3 [10,52]. In mice, *DUSP9* deficiency exacerbated the gain in body weight, liver weight, lipid accumulation and synthesis mediated by a high fat/high-cholesterol diet [34]. However, under the same dietary conditions, hepatocyte-specific *DUSP9* transgenic mice were less prone to accumulate lipids and triglycerides and to develop liver injury, fibrosis, inflammation, NAFLD and NASH [34]. Importantly, the hepatic over-expression of *DUSP9* directly affected the blood levels of glucose and insulin, glucose tolerance and the sensitivity of cells to insulin [34]. At a molecular level, *DUSP9* deficiency increased the phosphorylation of ERK1/2, JNK and p38, decreased the phosphorylation of IRS-1, AKT and GSK3-β proteins and increased the expression of pro-inflammatory cytokines and chemokines in an inhibitor of nuclear factor kappa B kinase subunit beta (IKK-β)/nuclear factor kappa B (NF-κB)-dependent manner. In contrast, over-expression of *DUSP9* reduced the phosphorylation of JNK and p38 MAPK in hepatocytes. Interestingly, these authors uncovered a new target of *DUSP9* named ASK1 (also known as MAP3K5 or MAPKKK5) and demonstrated the capacity of *DUSP9* to control the SAP response downstream p38 MAPK by dephosphorylating ASK1 and decreasing *MAPK kinase 4/7* expression. Moreover, they demonstrated the requirement of ASK1 activation in SAP signaling, lipid accumulation and induced-insulin resistance in hepatocytes [34]. Finally, a recent study reported that higher expression of *DUSP9* in placental cytotrophoblasts is associated with hyperglycemia in pregnant women suffering from type 2 diabetes mellitus compared to pregnant women not presenting it. This raises the question of the role of *DUSP9* as an effector of gestational insulin resistance [32].

## 5. DUSP9 and Cardiac Disease

A recent paper by Jiang and collaborators reported that *DUSP9* is implicated in the pathological process of heart failure [30]. The progression of this cardiac alteration was attributed to both pathological heart tissue remodeling and over-activation of stress responses, involving many signaling pathways including MAPKs. The authors showed that *DUSP9* is required to limit pressure overload-mediated cardiac hypertrophy and heart abnormality. The protective effect of *DUSP9* on cardiomyocytes was mediated by the binding and dephosphorylation of the MAPKKK ASK1, subsequently preventing the activation of the downstream p38/JNK pathway [30]. In absence of additional reports on the role of *DUSP9* in heart pathologies, future studies are required to improve our understanding of the relationship between *DUSP9* phosphatase and cardiac diseases.

## 6. DUSP9 in Cancers

*DUSP9* is implicated in many types of cancer (Table 2; Figure 3). It was identified by the Walktrap algorithm as an HCC-associated gene and a potential target for therapeutic research because of its interaction with MAPKs and the proto-oncogenes JUNB and FOSB [68]. However, its mechanistic role in cancer is unclear and seems context-dependent. In this section, we summarize the main data gathered from the literature on the role of *DUSP9* in cancer.

### 6.1. Breast Cancer

Despite progress in highly effective chemotherapies against breast cancer, the triple negative breast cancer (TNBC) subtype remains highly resistant to treatment and presents a high risk of recurrence and mortality [75]. A link between DUPS9 and breast cancer was first described in 2018 by Lu and collaborators who investigated MAPK pathway activity in response to chemotherapy in TNBC [17]. Treatment of MDA-MB-231 cells with paclitaxel or carboplatin inhibited ERK phosphorylation and induced p38 phosphorylation. These effects were lost by the genetic or pharmacological blockade of HIF1α, demonstrating their HIF1α dependency. Using a large cohort of breast cancer samples from the TGCA database, the authors found a strong positive correlation between *DUSP9* expression and a 10-gene HIF signature in a TBNC-enriched tumor subtype. Depletion of *DUSP9* in MDA-MB-231 cells abolished paclitaxel-mediated ERK inhibition, indicating that ERK is a substrate of *DUSP9*. MDA-MB-231 cells cultured as mammospheres, which promotes breast cancer stem cell (BCSC) enrichment, expressed a higher amount of *DUSP9* protein compared to cells cultured in standard monolayer condition. Additional analysis of chemotherapy-resistant BCSCs in mammospheres showed that paclitaxel or carboplatin induces *DUSP9* expression by promoting the direct binding of transcription factor HIF1α to *DUSP9* promoter and its transcriptional activation. This leads to ERK inhibition and p38 activation, thereby driving TNBC cells to switch from ERK to p38 signaling, evade chemotherapy and resist the drug through enrichment of BCSCs [17]. *DUSP9* is therefore a key factor in chemotherapy-induced BCSC enrichment. In parallel, ERK inhibition caused the upregulation of *NANOG* and *KLF4* mRNAs due to ERK-substrate FoxO3 dephosphorylation, while activated p38 indirectly stabilized *NANOG* and *KLF4* mRNA through ZFP36L1 RNA-binding protein following its phosphorylation by MK2 [17]. Expression of the pluripotency factors *NANOG*, *SOX2* and *KLF4* induced by paclitaxel in MDA-MB-231 cells was attenuated by *DUSP9* knockdown [17]. Therefore, *DUSP9* expression promotes BCSC phenotype specification by inhibiting the ERK pathway and supporting the p38 pathway (Figure 3). In vivo, inactivation of *DUSP9* did not significantly decrease the ability of initiating cells to form tumors. This result was consistent with in vitro results showing that *DUSP9* knockdown did not significantly decrease the number of mammosphere formations in the absence of paclitaxel [17]. Together, these data suggest that *DUSP9* contributes to cancer stemness and drug resistance and thus positively influences the growth and progression of TNBCs. This hypothesis, which is in line with the findings of another study [35], was further supported by an increase in *DUSP9* transcript levels in breast cancer patients who received chemotherapy [17]. However, *DUSP9* mRNA level was not an independent prognostic factor for relapse-free survival [17]. In 2020, Jimenez and collaborators showed that pERK1/2 is repressed and p38 is upregulated following engraftment of TNBC cells HCC70, HCC1806 and MDA-MB-48 in immune-deficient mice [35], The repression of pERK1/2 was associated with *DUSP9* upregulation and high phosphatase activity. The authors also found an inverse relationship between phospho-ERK1/2 activity and *DUSP9* expression in TNBC xenografts, as well as in TNBC mammospheres [35], However, in contradiction with the previous study, they showed that the pharmacological or genetic depletion of *DUSP9* reduced the growth of TNBC mammospheres in vitro, as well as tumor growth and mass in vivo [35],

Although data from two conflicting reports do not suggest the pro-tumorigenic role of *DUSP9* in TNBC, *DUSP9* regulates the ERK pathway in TNBC and plays a role in chemotherapeutic resistance and cancer stemness of TNBC cells (Table 2, Figure 3).

### 6.2. Colorectal Cancer

The first link between *DUSP9* and CRC was documented in 2006 by Sansom and collaborators. They showed an upregulation of *DUSP9* mRNA in *APC*-deficient mice intestines, suggesting a connection between *DUSP9*, the Wnt pathway and the early stage of colon tumorigenesis [76]. In 2015, a quantitative pyrosequencing methylation screening applied to 79 CRC and 22 healthy colon samples evidenced differences in the methylation status of the *DUSP9* promoter, although no conclusion could be drawn regarding the role of *DUSP9* promoter methylation in colon carcinogenesis [77]. Later, antagonistic interactions between ERK signaling and RAR signaling was reported in CRC cells [49]. Indeed, by increasing *DUSP9* expression, RAR signaling counteracts ERK pathway activation. As a result, ERK1/2 kinases were dephosphorylated and CRC cells engaged in differentiation. On the other hand, ERK signaling was shown to inhibit RAR signaling by a transcriptional mechanism mediated by RIP140/histone deacetylase (HDAC) [49]. The upregulation of *DUSP9* mediated by RAR signaling in CRC cells was consecutive to the direct binding of RAR to the *DUSP9* promoter through a consensus sequence for the RAR/RXR heterodimer corresponding to the DR1 element [49].

Analysis of *DUSP9* expression by different experimental approaches in paired CRC tissues showed the downregulation of *DUSP9* mRNA and protein in CRC tissues compared to peritumoral tissues [56]. Patients with low *DUSP9* transcripts had significantly shorter overall survival and recurrence-free survival than those with high protein levels. In addition, decreased *DUSP9* level was closely associated with larger tumors, deeper invasion, and advanced cancer stage, indicating that *DUSP9* down-regulation is associated with tumor progression and poor prognosis in CRC [56].

At a mechanistic level, the *DUSP9* promoter was hypomethylated in normal intestine compared to CRC, and the treatment of SW480 cells treated with 5-aza-dC, an inhibitor of DNA methyltransferase activity, induced *DUSP9* protein level [56]. Interestingly, hypermethylation of *DUSP9* promoter was also observed in many other solid tumors including bladder urothelial carcinoma, breast carcinoma, cervical squamous cell carcinoma, endocervical carcinoma, lung carcinoma, and pancreatic carcinoma. Therefore, hypermethylation of the CpG islands near the transcription initiation site of *DUSP9* promoter could in part explain *DUSP9* downregulation in CRC and other cancers [56]. In addition, Qiu and collaborators showed that miRNAs also participate in *DUSP9* downregulation in CRC. They showed that miR-1246 reduces *DUSP9* level in CRC cells through an miR-1246-binding site located in the 3′UTR of its transcript and blocks *DUSP9*-mediated phenotypical changes [56]. Moreover, miR-1246 levels were inversely correlated with *DUSP9* mRNA in 30 CRC samples.

At the functional level, forced *DUSP9* expression affected the proliferation, migration, invasion and epithelial–mesenchymal transition (EMT) of CRC cells in vitro and in vivo [56]. (Table 2; Figure 3). In vitro, changes in the growth and migration of CRC cells were cell-dependent [56]. For example, *DUSP9*-silenced SW480 cells produced larger tumors in mice compared to controls. In contrast, tumors in mice with *DUSP9*-expressing LoVo cells grew more slowly and their size was significantly smaller than in controls [56]. Transcriptome profiling studies performed in *DUSP9*-silenced SW480 cells reported an enrichment of several signaling pathways related to tumor growth and metastasis, such as ERK, JNK, WNT, AKT/mTOR and ERBB [56]. *DUSP9* silencing activated ERK signaling and promoted tumor development [49,56]. Treatment of *DUSP9*-silenced SW480 cells with PD98059 (a specific inhibitor of the ERK signaling pathway) counteracted their growth advantage, while *DUSP9*-expressing LoVo cells treated with curcumin (an activator of the ERK signaling pathway) grew significantly faster [56]. Together, these data show that *DUSP9* plays an important role in CRC cell tumorigenicity but its role depends heavily on the cellular context and likely on the functional balance between the ERK pathway and the p38 pathway.

### 6.3. Gastric Cancer

In gastric cancer (GC), the expression level of *DUSP9* is low at all stages of the disease and is the lowest in advanced stages [70]. This decreased expression starts very early in tumor progression. *DUSP9* down-regulation is detectable in adenomas and also observed in GC cell lines compared to normal gastric cells [70]. Bisulfite sequencing PCR analysis of 30 GCs and matched healthy tissues from gastric mucosa showed high methylation of the *DUSP9* promoter at the CpG island in cancerous tissues compared to normal tissues. Treatment of GC-derived MKN-1 cells with the DNA-hypomethylating 5-aza-2′-deoxycytidine (5-aza-dC) drug significantly induced *DUSP9* expression [70]. In agreement with another study [56], downregulation of *DUSP9* in GC is likely due to *DUSP9* promoter hypermethylation and transcription shutdown. Forced expression of *DUSP9* in MKN-1 cells inhibited cell growth by approximately 50%, while *DUSP9* silencing promoted cell growth by 74%. *DUSP9*-expressing MKN-1 cells remained blocked in the S-G2/M phases and presented a down-regulation of the cell cycle-associated proteins c-Jun, CCND1, CDK4 and CDK6, while *p21* expression was stimulated. These anti-proliferative effects were counteracted by the treatment of *DUSP9*-expressing cells with the JNK inhibitor SP600125. In vivo, mice xenografted with *DUSP9*-expressing MKN-1 cells developed larger tumors than the control mice [70].

Altogether, these data demonstrate that *DUSP9* acts as a tumor suppressor in GC cells in a JNK pathway-dependent manner (Table 2; Figure 3).

### 6.4. Liver Cancer

*DUSP9* also appears to play an important role in liver tumors (Table 2, Figure 3). The Walktrap algorithm identified *DUSP9* as one of the key genes involved in the onset of HCC [68], likely because of the importance of the ERK pathway in this hepatic malignancy [10,78]. However, its exact functions in liver cancer is a matter of debate [15,21,73].

In 2013, a first analysis of biological samples showed that *DUSP9* mRNA and protein were under-expressed in HCCs compared to non-tumoral livers [79]. Using the hepatoblastoma-derived and not the HCC-derived cell line HepG2 [74,80], the authors also showed that the depletion of *DUSP9* slightly increased cell proliferation and improved the basal survival of cells in vitro. Analysis of a cohort of 134 HCCs showed that the *DUSP9* decrease was a strong independent prognostic factor and was associated with poor overall survival and a high risk of recurrence [79]. The same team published a second paper in 2019 reporting the tumor suppressive role of *DUSP9* in liver cancer [15]. They first showed that *DUSP9* interacts with ERK1/2 kinases in tumor tissues and hepatoblastoma-derived HepG2 cells. *DUSP9* expression was decreased in the human liver tumor cell lines HepG2, SK-Hep1 and SMMC-7721 compared to the human hepatocyte cell line LO2. After the laborious selection of one effective anti-*DUSP9* small interfering RNA (one out of six tested), they showed that medium *DUSP9* silencing enhanced ERK1/2 phosphorylation and inversely, that a moderate increase in phosphatase level (50% increase) decreased ERK1/2 phosphorylation [15]. The depletion of *DUSP9* promoted the growth and clonogenicity of hepatoma cells, and the formation of spheroids, while forced *DUSP9* expression produced the opposite effect. Treatment of hepatoma cells with the ERK kinase antagonist PD98059 totally abrogated the proliferative effect induced by the depletion of *DUSP9*. Unfortunately, this inhibitor was not tested alone and the statistical test used to analyze these data was incorrect, precluding any definitive conclusions [15]. In vivo, forced expression of *DUSP9* inhibited the growth of grafted hepatoma cells in mice. Consequently, tumor volume and mass were significantly smaller compared to cells expressing basal or low level of *DUSP9* [15]. Inversely, knockdown of *DUSP9* promoted tumor development and expansion in mice. Additional analyses showed that the phenotypical changes mediated by *DUSP9* were due to enhanced and diminished ERK1/2 dephosphorylation in *DUSP9* silenced and *DUSP9*-expressing tumors, respectively. Together, these data supported the hypothesis that *DUSP9* exerts a tumor suppressive role in HCC by inactivating the ERK pathway and lowering the level of the ERK targets MYC and Cyclin D1 [15]. The authors then measured the level of *DUSP9*, ERK1/2 and phospho-ERK1/2 in eight HCC samples and found, as previously reported [79], that *DUSP9* was decreased in HCC, and that ERK1/2 and phospho-ERK1/2 were increased in the same samples. These data were confirmed by immunohistochemistry (IHC) analysis using a cohort of 160 HCCs. Finally authors correlated the expression of *DUSP9* with clinicopathological factors and found a significant association between low *DUSP9* expression or high ERK1/2 and phospho-ERK1/2 levels and poor overall survival [15].

In another study, again misusing the HepG2 cells as HCC-derived cells, the authors investigated at the cellular and molecular level the benefit of using phenolic compounds in metabolic syndromes [55]. These syndromes may evolve into an HCC [55]. Phenolic compounds enhanced glucose consumption and reduced the inflammatory response of palmitate-treated HepG2 cells. This effect was associated with dephosphorylation of p38 and JNK1, a decrease in miR-212 and a slight increase in *DUSP9* [55]. The authors also found that miR-212 could repress *DUSP9* by directly binding its 3′UTR. These data further support the role of *DUSP9* in the regulation of stress responses related to obesity and insulin resistance (see Section 4).

In 2008, *DUSP9* was reported to be part of a 16-gene signature discriminating hepatoblastomas classified as C1 and C2 groups [73]. Hepatoblastoma is an embryonal tumor and a primary liver cancer affecting children below the age of five. Cairo, Armengol and collaborators found that the *DUSP9* level was increased in most C2 samples, a group of tumors associated with poor prognosis, an advanced disease stage, high proliferation, chromosomal instability, early fetal origin and MYC signaling [73]. We further found an overexpression of *DUSP9* in the poor prognosis C2A group [74]. Therefore, unlike the previous studies [15,79], *DUSP9* may play rather an oncogenic role in a subgroup of hepatoblastoma (Table 2), and the early fetal origin of C2 tumors again supports a link between *DUSP9* and highly undifferentiated tumoral cells such as cancer stem cells or precursors.

In an attempt to clarify the role of oval cells (also known as intra-hepatic stem cells) in HCC onset, Sanders and collaborators investigated the role of mTOR1 and mTOR1 inhibitor rapamycin in liver cancer initiation using different liver injury models in Fisher rats [81]. Oval cells generated by the choline deficient plus ethionine model expressed high levels of the *DUSP9/MKP-4* gene and cyclin D1. This effect was counteracted by the administration of rapamycin. *DUSP9* levels were also higher in fetal livers than in adult livers. The authors concluded that *DUSP9* is a marker of fetal liver and is more specifically expressed by the very undifferentiated oval cells [81], which is in agreement with the increased expression of *DUSP9* in embryonic-derived hepatoblastoma [73,74].

A very recent report from a Canadian team also investigated the role of *DUSP9* in HCC [21]. By interrogating GEO datasets, the authors found that *DUSP9* transcript (along with Glypican-3 and α-fetoprotein) was overexpressed in HCCs, fetal livers and human hepatoma cell lines compared to their corresponding non-tumoral counterparts. Higher *DUSP9* expression was associated with an increased amount of serum α-fetoprotein, more advanced disease stage, microvascular invasion and higher risk of recurrence after surgery, but not with overall survival. *DUSP9* protein was also more expressed in hepatoma cell lines than in immortalized hepatic THLE-2 cells, and in HCC samples than in non-tumoral livers [21]. *DUSP9* upregulation in HCC was associated with transcriptional regulation mediated by ETS transcription factors. Moreover, *DUSP9* mRNA expression positively correlated with expression of *ETV4*, *ELF3*, *ERF* and *ETV5* transcripts. In opposition with a previous study [15], Chen and collaborators showed that the forced expression of *DUSP9* increased the proliferative capacity of hepatoma cells expressing low level of *DUSP9* protein. Inversely, hepatoma cell knockout (KO) for *DUSP9* proliferated less, had reduced clonogenic capacity, were more sensitive to doxorubicin and harbored higher ERK pathway activity [21]. In vivo, *DUSP9* silenced cells generated smaller tumors and had lower growing capacity. Finally, grafted *DUSP9*-KO Hep3B cells were unable to form a tumor in comparison with parental cells. *DUSP9* silencing was associated with increased MAPK activity and cell survival [21].

Altogether, these data show that the precise role of *DUSP9* in liver cancers is still a matter of debate and that additional investigations need to be carried out to determine if *DUSP9* acts as an oncogene or a tumor suppressor in liver malignancies.

### 6.5. Lung Cancer

A transcriptome sequencing analysis in patients with LA identified *DUSP9* down-regulation as one factor associated with cell invasion and metastasis [72] (Table 2; Figure 3). Concerning the mechanism, the blockade of miR-1233-3p function by the circular RNA has_circ_0004050 stimulated *DUSP9* expression in A549 cells leading to ERK/JNK pathway inactivation and tumor inhibition. This suggested that *DUSP9* acts as a tumor suppressor in lung cancer [82].

### 6.6. Kidney Cancer

The role of *DUSP9* in renal cancer has been highlighted in several studies and *DUSP9* is regularly found to be down-regulated in renal tumors compared to normal tissues [33,83,84]. In 2010, Zhou and collaborators analyzed 10 clear cell renal cell carcinomas (ccRCC) and matched normal adjacent tissues. They reported for the first time the decreased expression of *DUSP9* mRNA in tumor tissues, suggesting the pivotal role of *DUSP9* in the development of renal cancer [84]. One year later, the down-regulation of *DUSP9* mRNA and protein was further confirmed in ccRCC using real time quantitative RT-PCR and IHC [33]. Additional analysis in a cohort of 211 ccRCC samples showed that the *DUSP9* decrease was associated with poor overall survival in patients and higher pathological disease stages (Fuhrman grade) [33]. In 2020, a new study reported the decreased expression of *DUSP9* mRNA in approximately 81% of tumor tissues and most ccRCC cell lines [69]. The decrease in *DUSP9* mRNA was always accompanied by a decrease in *DUSP9* protein [33,69]. The same year, the down-regulation of *DUSP9* mRNA in ccRCC was confirmed by a Polish group [83]. Phenotypically, forced expression of *DUSP9* in ccRCC cell lines inhibited cell growth and migration in vitro and impeded tumor growth and size in vivo in a xenograft nude mouse model, thus showing for the first time the tumor suppressor role of *DUSP9* in renal cancer [69]. This tumor suppressor role was mediated by inhibition of the mTOR pathway, as shown by dephosphorylation of mTOR on serine 2448 and a decrease in its pathway-associated targets HIF-1α, Sox2 and c-Myc (Figure 3) [69].

### 6.7. Skin Cancer

Using a carcinogenic model of squamous cell carcinoma (SCC) and a transcriptomic approach, Liu and collaborators showed that *DUSP9* mRNA is down-regulated in tumor initiated cells compared to non-transformed parental cells, and completely lost in malignant cells, which harbor the typical characteristics of advanced-stage disease [71]. Carcinogenic induction using either UVB light or 7,12-dimethylbenz (a) anthracene compound led to a loss of *DUSP9* expression in 100% and 40% of treated mice, respectively [71]. Using a lentiviral approach, forced expression of *DUSP9* induced the death of nearly 90% of initiated cells after nine days by inhibiting the phosphorylation of ERK, JNK and p38. Under the same conditions, no phenotypical change was observed with *DUSP6/MKP3* ectopic expression. Unexpectedly, *DUSP9*-expressing cells accumulated in the cell-death associated G2/M phase and were not arrested in the G1 phase, as is normally the case following ERK, JNK and p38 dephosphorylation. Cells in the G2/M phase were larger with condensed chromatin and underwent a mitotic catastrophe. Additional analysis showed that *DUSP9* disorganized the microtubules [71]. In newborn BALB/c mice, the graft of *DUSP9*-expressing initiated cells resulted, on average, in seven-fold smaller tumors than the control cells 30 days after inoculation [71]. Histological analysis showed that tumors resulting from *DUSP9*-transduced cells did not express *DUSP9*, suggesting that engrafted cells at the origin of the tumor were likely not infected. Similar in vivo data were obtained with cells conditionally expressing *DUSP9* following induction with tetracycline. Histological analysis of the tumor tissues showed a major change in the morphology of cells, which were larger and presented decondensed nuclei, indicating a tumoricidal effect (Table 2, Figure 3). Altogether, these in vitro and in vivo data suggest that *DUSP9* plays a role in preventing SCC cell transformation and skin carcinoma expansion.

## 7. DUSP9 Is the Target for Therapy

Given the central role of *DUSP9* in cancer and metabolic diseases, some groups have attempted to identify potent molecules specifically targeting it. Unfortunately, many human protein tyrosine phosphatases share similarities in their active site [43,85,86], thus precluding the development of specific selective drugs targeting the catalytic core of *DUSP9*.

A first attempt was to manipulate the activity of *DUSP9* by targeting the loops surrounding the active phosphatase site and its spatial conformation to increase its efficiency (Figure 1C). To this end, 85,000 compounds with the physicochemical characteristics of drug candidates were selected from the InterBioScreen library and tested by virtual screening using the three-dimensional structure of the *DUSP9* active site determined by crystallographic studies and in silico interaction simulations [43]. The 100 molecules with the best score in the virtual screening were selected. The study of the inhibitory activity of these molecules on *DUSP9*, carried out in vitro using a recombinant human protein, selected five very powerful molecules. Analysis of the molecular mechanism showed that four of the five compounds bound to *DUSP9* protein between the catalytic core and the surrounding loops with a higher affinity near the loops. This interaction increased the substrate affinity and specificity of *DUSP9* [43]. The loops which surround the catalytic site and participate in the interaction represent a good therapeutic target (Figure 1). The structure of these loops is different from one phosphatase to another, and they are less hydrophilic than the active site. Targeting these loops could therefore lead to the development of specific drugs capable of blocking the movement of loops in the active conformation of *DUSP9* [43].

A recent study reported the capacity of the lncRNA *LincU* to maintain *DUSP9* in an active state and block the MAPK pathway [15,48]. This remarkable result suggests that a therapeutic approach aiming at keeping *DUSP9* in an active state may be beneficial for the treatment of patients with metabolic syndrome or tumor. This could be particularly useful for the development of specific treatments against type 2 diabetes and tumors under-expressing *DUSP9* [43,67,71].

Another study reported that ginkgolic acid, which is extracted from the *Ginkgo bilola* leaf or seed coat, can inhibit *DUSP9* activity [87]. It was selected following the screening of 658 natural compounds by an enzymatic test based on a recombinant purified human *DUSP9* protein. In vitro, ginkgolic acid reduced *DUSP9* phosphatase activity by 70% [87]. Interestingly, ginkgolic acid was found to be a potent antitumor agent in several solid tumors, suggesting its therapeutic potential for the treatment of cancer [84,88,89] This suggests its putative use in tumors overexpressing *DUSP9* such as C2 or C2A hepatoblastomas. However, its use in clinical settings will depend on both the development of synthetic derivatives of the natural compound and the demonstration of their harmlessness in preclinical tumor models.

## 8. Conclusions

(1)*DUSP9* is tightly regulated at transcriptional, post-transcriptional and post-translational levels by transcription factors, promoter methylation, miRNAs, lncRNA and ubiquitination. Its rigorous regulation is therefore necessary to maintain normal cell function and physiological homeostasis.(2)*DUSP9* has a central role in sex differences, metabolic disorders and tumorigenesis. At a pathophysiological level, *DUSP9* is strongly involved in the regulation of insulin signaling, and consequently, of the downstream phosphorylation cascades and metabolic processes. Therefore, any therapeutic intervention to increase *DUSP9* expression or to control the activity of ERK1/2, JNK, p38 MAPK and/or ASK1 kinases could be beneficial for patients presenting metabolic syndromes such as type 2 diabetes, morbid obesity, liver cirrhosis, NAFLD or its most severe form, NASH.(3)*DUSP9* is central in tumorigenesis and is involved in many adult and pediatric cancers. It clearly acts as a tumor suppressor in kidney cancer, gastric cancer, skin cancer, CRC and lung cancer. Its high expression is associated with poor prognosis in C2A hepatoblastomas and with cancer stemness and drug resistance in breast cancer (Table 2; Figure 3). However, the role of *DUSP9* in the adult liver is still a matter of debate and needs further investigations to determine its pro- or anti-tumoral function in hepatic malignancies (Table 2; Figure 3). As NASH can lead to malignant HCC, therapies influencing *DUSP9* activity could also be beneficial in patients with liver cancer.(4)*DUSP9* is also an important gene in heart tissue preservation and can counteract the negative effect mediated by pressure overload on cardiac hypertrophy and cardiomyocytes.(5)Collectively, these data clearly demonstrated the critical role of *DUSP9* in both cell physiology and pathologies. It is therefore a promising therapeutic target to fight against the frequent human diseases that are diabetes, heart failure and cancers.

## Figures and Tables

**Figure 1 ijms-22-11538-f001:**
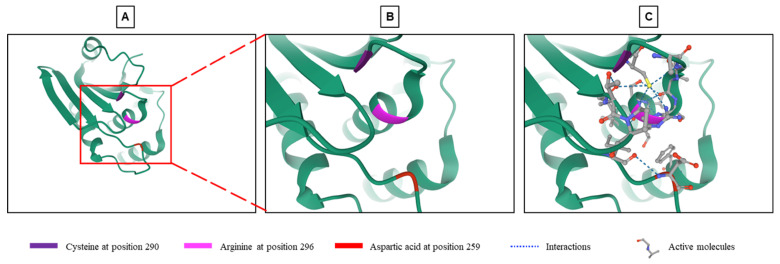
Structure of the phosphatase active site of *DUSP9*. (**A**) Crystallographic representation of the phosphatase site of *DUSP9* with a focus on its catalytic part (see region of interest in red box) composed of a cysteine at position 290, an arginine at position 296 and an aspartic acid at position 259 (from RCSB Protein Data Bank: https://www.rcsb.org/3d-view/3LJ8, accessed on 1 September 2021) [45]. (**B**) Higher magnification of region of interest shown in panel A. (**C**) Same image than in Panel B with inserted substrates and molecular interactions between the three amino acyls and substrate.

**Figure 2 ijms-22-11538-f002:**
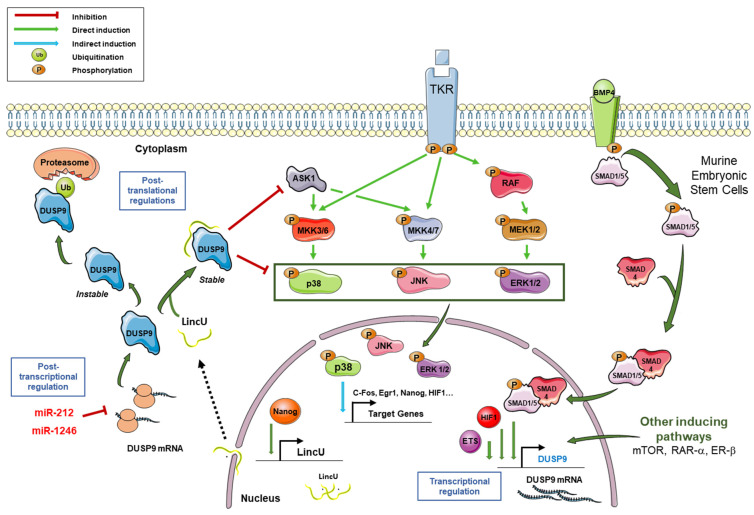
Regulation of *DUSP9* expression and connection with MAPK pathways. Following the activation of tyrosine kinase receptors (TKR), MAPK pathways are activated through the successive phosphorylations of MAPKKKs (among which ASK1), MAPKKs MKK3/6, MKK4/7 and MEK1/2, and MAPKs p38, JNK and ERK. Phosphorylated MAPKs translocate to the nucleus and induce expression of the downstream targets *c-FOS*, *ERG1*, *NANOG* and *HIF1*, among others. In murine embryonic stem cells, the binding of BMP4 on its receptor induces the phosphorylation of Smad1/5, which then associates with Smad4. The Smad1/5–Smad4 complex translocates to the nucleus, binds the *DUSP9* promoter and induces its transcriptional expression. Transcription factors *HIF1*-α and ETS can also induce *DUSP9* expression. Besides the BMP signal, mTOR, RAR-α and ERβ pathways can also potentiate the transcription of the *DUSP9* gene. Following *NANOG* binding on *LINCU* promoter, the long non-coding RNA *LincU* is transcribed and exported in the cytoplasm where it associates with *DUSP9* and stabilizes it. Stable *DUSP9* can dephosphorylate its substrates, including ERK1/2, p38, JNK and ASK1. In the absence of *LincU* RNA, *DUSP9* protein is unstable and is polyubiquitinylated before its degradation by the proteasome. *DUSP9* can also be post-transcriptionally regulated by miR-212 and miR-1246, which target the 3′-untranslated region of its mRNA. In summary, *DUSP9* expression is tightly regulated by transcriptional, post-transcriptional and post-translational mechanisms.

**Figure 3 ijms-22-11538-f003:**
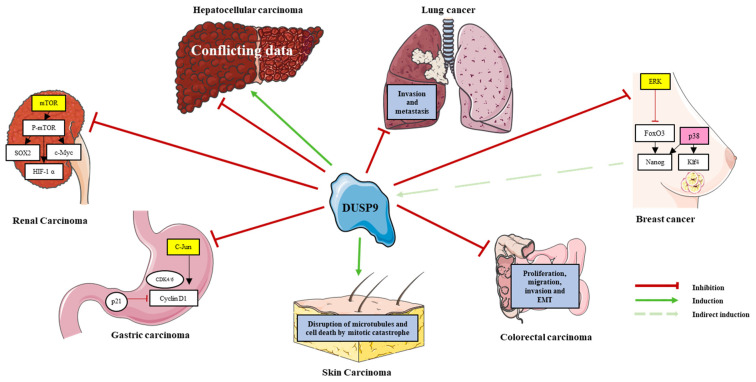
Role of *DUSP9* in various cancers. Positive and negative effect of *DUSP9* (see legend at the bottom right) on putative molecular pathways and oncogenic processes in the corresponding cancer. Targets of *DUSP9* phosphatase (shown in yellow boxes) are inactivated, and this inactivation induces p38 kinase activity (shown in pink box) in breast cancer. See Section 6 for a fuller description.

**Table 1 ijms-22-11538-t001:** General information and main functions of MAP kinase phosphatases (MKP).

Classification	Gene Symbol	Synonyms	Chromosomal Localization	Cell Localization	MAPK Substrates (Others)	Inducible by MAPKs	Main Functions in Physiological and Pathophysiological States
Typical MKPs	*DUSP1*	MKP1	5	Nuclear	JNK, p38 > ERK	ERK, p38	Involved in infectious diseases, pulmonary diseases, inflammatory disorders, atherosclerosis, tumorigenesis and tumor progression [22].
	*DUSP2*	PAC1	2	Nuclear	ERK, JNK, p38	ERK, JNK	Involved in immune and inflammatory responses, cancer, CLN3 disease and endometriosis [23].
	*DUSP4*	MKP2	8	Nuclear	ERK, JNK > p38	ERK	Involved in inflammatory cytokine secretion, susceptibility to sepsis shock, and resistance to *Leishmania mexicana* infection [24,25].
	*DUSP5*	hVH3	10	Nuclear	ERK	ERK	Plays an anti-inflammatory role and has tumor suppressive functions in several types of cancer [26].
	*DUSP6*	MKP3	12	Cytoplasmic	ERK	ERK	Plays a role in carcinogenesis in several cancers as an oncogene or a tumor suppressor [27].
	*DUSP7*	MKPX	3	Cytoplasmic	ERK, JNK, p38	N/D	Involved in some cancers [28].
	*DUSP8*	hVH5	11	Dually-located	ERK, JNK, p38	N/D	Plays a role in the central nervous system, circulatory system, urinary system, immune system, genetic diseases and cancers [29].
	** *DUSP9* **	**MKP4**	**X**	**Cytoplasmic**	**ERK >> p38, JNK**	**N/D**	**Involved in development of cardiac dystrophy, metabolic diseases and cancers** [30,31,32,33,34].
					**(MAP3K5/ASK1)**		
	*DUSP10*	MKP5	1	Dually-located	JNK, p38 >> ERK	N/D	Involved in immune response, anti-inflammatory response and some cancers [35].
	*DUSP16*	MKP7	12	Dually-located	JNK	N/D	Involved in non-alcoholic steatohepatitis and some cancers [36].
Atypical MKPs	*DUSP14*	MKP6	17	Dually-located	ERK, JNK, p38	N/D	Involved in immune response, bone diseases and cancers [37].
	*DUSP26*	MKP8	8	Nuclear	p38	N/D	Regulates neuronal cell proliferation and acts as an oncogene or a tumor suppressor depending on the cellular context [38].

Data were also extracted from the following publications: [3,11,14,16,39,40,41,42]. N/D: not determinate.

**Table 2 ijms-22-11538-t002:** Expression of *DUSP9* in cancers versus non-tumoral tissues and main results from the literature.

Organ	Cancer	Expression	Role	Main Results
Kidneys	Clear cell renal carcinoma	Low	Tumor suppressor	*DUSP9* blocks the growth and migration of clear cell renal cell carcinoma cells in vitro and tumor development in mice through mTOR pathway inhibition [69].
Stomach	Gastric carcinoma	Low	Tumor suppressor	*DUSP9* inhibits growth of MKN-1 cells through cell cycle arrest in S-G2/M phases and JNK pathway inactivation [70].
Skin	Skin carcinoma	Low	Tumor suppressor	*DUSP9* triggers tumoral cell death by arresting cells in G2/M phase and by inducing microtubule disruption and mitotic catastrophe [11,71].
Colon	Colorectal carcinoma	Low	Tumor suppressor	In vitro and in vivo, *DUSP9* affects proliferation, migration, invasion and epithelial-mesenchymal transition of colorectal cancer cells [56].
Lungs	Lung cancer	Low	Tumor suppressor	*DUSP9* down expression is associated with tumor progression, invasion and metastasis [72].
Breast	Breast cancer	Low	Involved in chemoresistance	*DUSP9* promotes cancer stem cell enrichment and chemotherapy resistance of triple negative breast cancer through HIF1α [17,35].
Liver	Hepatocellular carcinoma	Low or High	Oncogene or Tumor suppressor	Conflicting results report a tumor suppressive or an oncogenic role of *DUSP9* in adult liver cancer [15,21].
	Hepatoblastoma	High	Associated with poor prognosis	*DUSP9* is increased in poor prognosis C2 or C2A group [73,74].

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
