# Peer review of "DUSP9, a Dual-Specificity Phosphatase with a Key Role in Cell Biology and Human Diseases"

_ijms, 2021, doi:10.3390/ijms222111538_

Round 1
Reviewer 1 Report
The review article by Khoubai and Grosset discussed the role of DUSP9, a threonine-tyrosine dual-specificity phosphatase in cell biology and human diseases. The authors did a nice job summarizing recent data on a cellular phosphatase that is important in human pathologies and a potential target in therapy for diabetes, cancers and other diseases. Although the length of the review can be shortened in many areas with different writing style, the current version of the manuscript is readable and understandable to the readers.
Comments:
- Although this is not important for the final publication, but the line numbers in the manuscript during the review process should be continuous. It would be easier to follow for the reviewers and comment on specific lines.
- The title of the manuscript should use only DUSP9 and remove MKP-4 as to mention the dual phosphatase. DUSP9 was mostly used throughout the manuscript and is more common than MKP-4.
- Please use the same format for all protein and gene name. For example, ERK1/2 was presented as Erk1/2 in few instances (Page 13 Line 10). Dusp9 should be replaced with DUSP9 on Page 17 Line 19.
- In many places, the authors used the term DUSP9 expression. The protein does not express, the gene expressed. Therefore, while mentioning expression, the authors should use the gene name instead of protein.
- The sub-sections; DUSP9 in cardiac disease, DUSP9 in cancers, and DUSP9 in therapy can be presented as individual sections.
- DUSP9 and cardiac disease section seems very short and abruptly ended. Few more references if available should be added to discuss this section. If no additional study has been done, then the authors should mention in the paragraph and give the rationale of studying DUSP9 in cardiac disease in future.
- The title for the sub-section III ‘DUSP9 in therapy’ should be changed. For the therapy, DUSP9 is not being used. Rather, DUSP9 is the target for therapy.
- Page 5 Line 9: Use a comma (,) after ‘According to the PhosphoSite database’.
- Page 5 Line 16: Specify ‘It’.
- Page 7 Line 6: This information is redundant here as it is already mentioned in Page 6 Line 20.
- Page 13 Line 7: ‘Mek kinase’ should be replace with ‘MEK kinase’.
- Page 14 Line 11: Place ‘and’ between X chromosomal gene and is involved.
- Page 16 Lines 5-7: Rewrite this sentence or split into two separate sentences.
- Page 20 Line 12: ‘In vivo’ should be italic.
- Page 22 Line 11: Redundant word used. Remove the word ‘also’ after ‘Qiu and collaborators’.
- Page 23 Line 6: Use ‘the expression level of DUSP9’.
- Page 25 Line 14-16: This sentence is long and should break down to two sentences.
- Page 31 Line 2: Fix the typo. The sentence should be ‘DUSP9 is tightly regulated….’
- Page 31 Line 25: Heart feature or heart diseases?
- Figure 3: What are those empty dotted boxes on the line between Breast Cancer and DUPS9 refer to? A short description of the figure under the figure legends is suggested. The readers can check the ‘DUSP9 in cancers’ section for the full details.
- Table 1: The texts in the far-right column ‘Major functions in physiological and pathophysiological states’ are not aligned to the texts in the other columns. It would be easier to follow if the authors give some space between the rows or have lines for the clarity.
- Table 2: Replace the arrows for the expression with either writing low or high. The column organ can be removed or replace with the name of the organ.
Reviewer 2 Report
This is interest manuscript.
The authors discribed the normal and pathological functions of dual-specificity phosphatase 9 (DUSP9)/MAP kinase phosphatases-4 (MKP-4). The authors demonstrated data of the critical role of DUSP9 in both cell physiology and pathologies. Need to correct English.
The practic aspect of paper can be improve, because the demonstration of harmlessness of specific selective drugs targeting the catalytic core of DUSP9 is important.
